# Active conformation of the p97-p47 unfoldase complex

Yang Xu[1,3], Han Han[1,3], Ian Cooney[1], Yuxuan Guo [1], Noah G. Moran[2], Nathan R. Zuniga[2], John C. Price[2], Christopher P. Hill [1✉] & Peter S. Shen [1✉]

The p97 AAA+ATPase is an essential and abundant regulator of protein homeostasis that plays a central role in unfolding ubiquitylated substrates. Here we report two cryo-EM structures of human p97 in complex with its p47 adaptor. One of the conformations is six-fold symmetric, corresponds to previously reported structures of p97, and lacks bound substrate. The other structure adopts a helical conformation, displays substrate running in an extended conformation through the pore of the p97 hexamer, and resembles structures reported for other AAA unfoldases. These findings support the model that p97 utilizes a "hand-over-hand" mechanism in which two residues of the substrate are translocated for hydrolysis of two ATPs, one in each of the two p97 AAA ATPase rings. Proteomics analysis supports the model that one p97 complex can bind multiple substrate adaptors or binding partners, and can process substrates with multiple types of ubiquitin modification.

[1] Department of Biochemistry, 15 N. Medical Drive East, University of Utah, Salt Lake City, UT 84112, USA. [2] Department of Chemistry and Biochemistry, C100 BNSN, Brigham Young University, Provo, UT 84602, USA. [3] These authors contributed equally: Yang Xu, Han Han. ✉email: chris@biochem.utah.edu; peter.shen@biochem.utah.edu

AAA unfoldases, including the abundant and essential protein p97 (also known as VCP), comprise a major subset of the large family of AAA+ (ATPases Associated with diverse cellular activities) enzymes that assemble as hexamers to unfold substrate proteins. p97 unfolds client proteins in multiple cellular pathways that include protein degradation, cell cycle progression, genomic stability, and membrane trafficking[1]. Its central role in protein homeostasis has made it an attractive target for anticancer[2] and antiviral[3] therapeutics, and missense mutations are causative of multiple degenerative diseases, including multisystem proteinopathy-1, Charcot-Marie-Tooth Type 2, and familial amyotrophic lateral sclerosis[4]. This has motivated structural studies, which include high-resolution crystal and cryo-EM structures of substrate-free complexes[5,6]. These substrate-free p97 structures display six-fold rotational symmetry, with the D1 and D2 ATPase cassettes in each subunit forming stacked rings. The N-terminal domains (N) are also usually seen in a six-fold symmetric arrangement, adopting a 'down' conformation in the presence of ADP and a less ordered 'up' conformation in the presence of an ATP analog. In keeping with these published structures, the mechanism of p97 substrate processing has been postulated to occur by toggling between these up/down, ATP/ADP states[7].

In recent years, multiple structures of related AAA unfoldases have been reported in complex with peptides that are believed to mimic substrate bound in a translocating/unfolding conformation[8]. Notably, in contrast to the previously reported six-fold symmetric structures of p97, these peptide-bound AAA unfoldase structures revealed an asymmetric conformation in which at least four and as many as all six subunits adopt a helical configuration that binds substrate peptide. In these structures, subunits are typically seen to bind ATP at the subunit interfaces that define the helical configuration, while ATPase sites of the other subunits typically appear to bind ADP or lack bound nucleotide. A similar conformation was seen for VAT, an archaeal homolog of p97/Cdc48, apparently visualized in the act of unfolding a neighboring VAT[9]. Moreover, since the submission of this manuscript, a similar conformation was reported for p97 bearing both a pathogenic mutation and an ATPase site mutation when reconstituted in complex with model substrate and the Ufd1/Npl4 cofactor[10]. These assemblies differ from the rotationally symmetric p97 structures, and are generally thought to indicate a mechanism in which ATPase sites fire sequentially around the hexamer ring and subunits move "hand-over-hand" along the substrate polypeptide to pull/translocate the substrate through the hexamer pore and thereby drive unfolding.

We set out to visualize the conformation of native p97, lacking any mutation, in its authentic state. This was achieved by affinity purification from human tissue culture lysates using a recombinant adaptor protein in the presence of the flexible ATP/ADP analog ADP·BeF$_x$. Subsequent structure determination by cryo-EM revealed two conformations, one resembling the rotationally symmetric state that lacks bound substrate, the other adopting the helical/spiral conformation with substrate apparent in the translocation pore. These structures are consistent with the model that authentic cellular p97 partitions between the inactive symmetric conformation and the substrate engaged asymmetric conformation, and employs the canonical hand-over-hand mechanism to translocate and unfold substrate.

## Results

**Sample preparation and structure determination.** Native p97 complexes were isolated from human tissue culture lysates by affinity purification of the substrate adaptor p47 (Fig. 1a, Supplementary Fig. 1). Recombinant FLAG-tagged p47 was expressed

in bacteria, purified, and added to lysate derived from HEK293S cell treated with the proteasome inhibitor bortezomib. p97 was subsequently co-purified with p47 by co-immunoprecipitation (co-IP) in the presence of the non-hydrolysable ATP analog ADP·BeF$_x$ and used for single particle cryo-EM reconstruction (Supplementary Figs. 2–4, Supplementary Table 1). Particle classification revealed two major states. One state displayed six-fold rotational symmetry that is superimposable with previously reported structures of p97 in the ADP-bound state[5,6] (Supplementary Fig. 3). The other state showed p97 in an asymmetric configuration that resembles the substrate-bound conformation of other AAA unfoldases.

**Reconstruction of symmetric, substrate-free p97.** The symmetric reconstruction does not display bound substrate and presumably represents an idle state of the complex. In this conformation, both D1 and D2 rings are stacked in planar arrangements. The N domains adopt a 'down' conformation that is co-planar with the D1 ring. This is consistent with existing structures of p97 in an ADP-bound state[6], although the resolution of our reconstruction does not permit nucleotide assignment. The periphery of the N-domains contains additional densities that are consistent with the UBX domain of p47[5] (Supplementary Fig. 3F). Focused classification over each of the six N-domains revealed UBX occupancies between 59-93%, suggesting that most, but not all, of the potential UBX-binding sites are engaged with the adaptor (Supplementary Fig. 3G, H). Although the N-domains are not saturated with UBX binding, we cannot rule out the possibility that some p47 may have dissociated from p97 between protein isolation and cryo-EM grid preparation. Nevertheless, our results are consistent with a recent report that reconstituted p47–p97 complexes display variable levels of UBX occupancy[11].

**Reconstruction of asymmetric, substrate-bound p97.** The asymmetric reconstruction displayed five of the six p97 protomers in a helical arrangement (Fig. 1b, c, Supplementary Fig. 4) and, in contrast to the symmetric reconstruction, displays a strand of what is presumably substrate polypeptide extending through the entire pore of the hexamer. The five helical subunits (A–E) are well-ordered in both D1 and D2 rings. In contrast, subunit F is pulled away from the peptide-binding groove, makes relatively open interfaces with its neighboring A and E subunits, and has poorly resolved density (Fig. 1b). For both D1 and D2, the helical interfaces are stabilized by binding of ADP·BeF$_x$ (ATP) at active sites that, as expected, are formed primarily by residues of the first subunit and completed by a pair of arginine residues from the second subunit (Fig. 2, Supplementary Fig. 5). Density at the active site of subunit E is most consistent with ADP, while density at the active site of subunit F is too weak to visualize whether or not nucleotide is bound.

Like other members of the classical AAA clade of proteins, p97 contains a conserved intersubunit signaling (ISS) motif, which has been proposed to transmit information about the nucleotide status between adjacent subunits and to the pore loops[12]. Specifically, D1 L335 and D2 M611, which appear to be the key residues from the ISS in each domain, interact with the hydrophobic residues near the Walker B motifs of the neighboring subunit (Supplementary Fig. 6). This ISS structure is well conserved in the subunits that form a helix in the asymmetric substrate-engaged reconstruction of p97, and are closely superimposable with the equivalent regions from YME1[12], Vps4[13–17], Cdc48[18,19], and human Spastin[20], but are somewhat divergent in *Drosophila* Spastin[21].

All six N domains adopt the 'up' position (Fig. 1c, d). Previous reports indicate the 'up' position is favored by ATP binding[4]. This

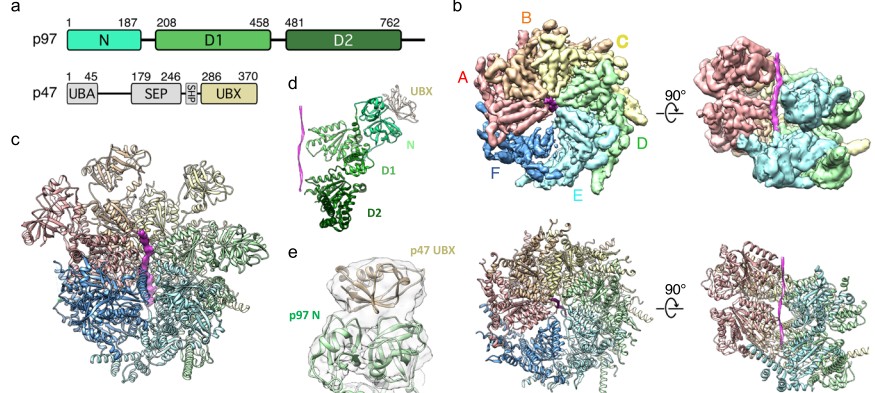

**Fig. 1 Structure of substrate-bound p97–p47 complex. a** Domain organization of p97 and p47. Domains are labeled and residue ranges indicated. **b** Top and side views of high-threshold segmented map and refined model (subunits A-F of p97 labeled). The substrate peptide density and model are colored magenta. **c** Model of the complete p97 complex including N domains modeled in the 'up' conformation. **d** Model of a p97 protomer (subunit D shown) with the associated p47 UBX domain. Orientation corresponds to the side view on panel (**b**). **e** Low threshold density fitted with a p97 N domain and p47 UBX domain.

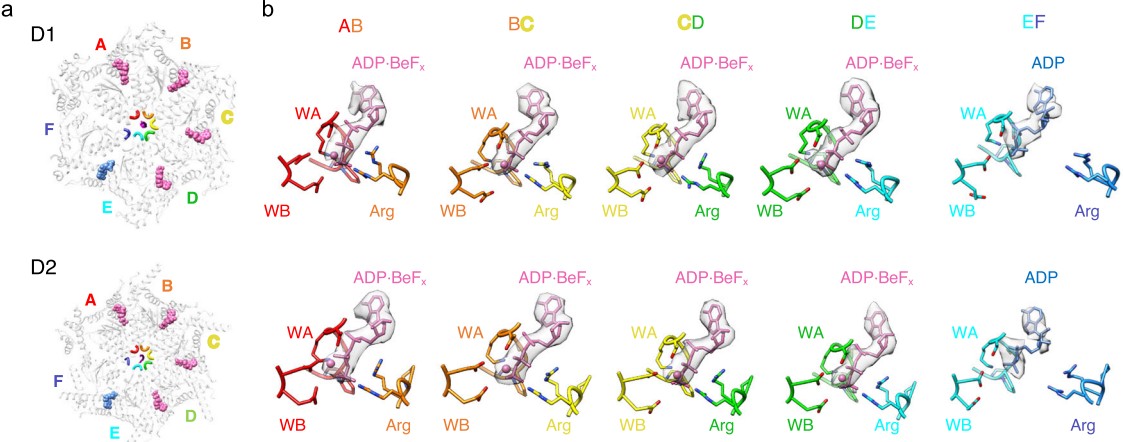

**Fig. 2 Nucleotide binding pockets. a** Top view of substrate-bound p97 hexamer with nucleotide models at subunit interfaces (pink spheres, ADP·BeF$_x$; blue spheres, ADP). Top row, D1 ring; bottom row, D2 ring. **b** Closeup views of nucleotide-binding pocket motifs with nucleotide density and model.

is consistent with the helical configuration being incompatible with intersubunit interactions that are formed by the N domain in the planar conformation. Although N domain densities are at the low local resolution, presumably because of mobility, low threshold views indicate the presence of p47 UBX domain density on all six of the N domains in the same manner as seen in a crystal structure of p97 N-D1 domains complexed with p47 UBX domains (Fig. 1e)[5]. Focused 3D classification over individual N-domains revealed UBX occupancies of 14-22% (Supplementary Fig. 7). The discrepancy in UBX occupancies between the 'up' and 'down' conformations may reflect different stoichiometries of p47 binding when the complex is active or idle. We also note the possibility that the increased mobility of the N domain in the 'up' position may weaken potential UBX densities and reduce the sensitivity of focused 3D classification.

**Substrate binding**. The substrate-bound subunits (A–E) are related to each other by an ~60° rotation and ~6.5 Å translation. This matches the helical symmetry of dipeptides in a canonical β-strand conformation. Correspondingly, the peptide density is fit well by a β-strand, with successive substrate dipeptides making

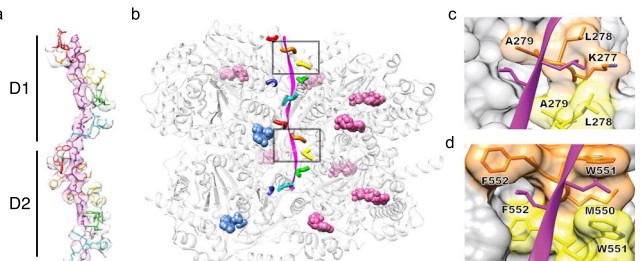

**Fig. 3 Interactions between p97 and substrate in the central pore. a** Side view of density and model of substrate (magenta) and pore loop 1 (subunit colors). **b** Side view of the p97 hexamer (ribbon) with pore loop 1 (subunit colors) and ATP (pink spheres) or ADP (blue spheres). Dashed boxes indicate closeup views in panels (**c**) and (**d**). **c** Closeup of the model showing the D1 substrate dipeptide binding units at the BC interface. The three other substrate-binding D1 subunit interfaces are very similar (i.e., AB, CD, and DE). Pore loop 1 residues are labeled. Substrate residues are modeled as leucine for clarity. **d** Equivalent to panel c for a substrate dipeptide binding in the D2 pore. Equivalent binding is observed at the other substrate-binding D2 interfaces.

equivalent interactions with successive p97 subunits through both the D1 and D2 pores (Fig. 3a, b). Pore loop 1 and pore loop 2 residues of both D1 and D2 contribute to substrate binding, with the tightest interactions apparently formed by the pore loop 1 residues Leu278-Ala279 from D1 and Trp551-Phe552 from D2 (Fig. 3c, d). These residues correspond to a canonical aromatic-hydrophobic motif in AAA+ unfoldases, and their side chains form a series of notches that can accommodate a wide variety of substrate side chains, thereby explaining the ability of p97 and other AAA unfoldases to bind and translocate many different protein substrates[22].

Interestingly, substrate density is weaker in D1 than it is in D2, which implies a weaker interaction with the D1 pore loops. This is consistent with the identity of D2 residues Trp551 and Phe552, which seem ideal for the formation deep notches that, in the highly solvated context of the p97 pore, can accommodate essentially any substrate side chain. By contrast, the equivalent D1 residues, Leu278 and Ala279, create shallower binding pockets for the substrate side chains. Consequently, consistent with the weaker density, substrate is likely to be bound less tightly in D1 compared to D2. Weaker interactions in D1 appear to be conserved between yeast and human[18], and to be important for biological function because substitutions of the orthologous Leu278 to Phe, Trp, or Tyr causes a lethal phenotype in yeast[23].

**Comparison to other AAA unfoldases**. To visualize the similarity in substrate coordination between p97 and other AAA unfoldases, 14 structures (all determined in the past 5 years) were superimposed on the pore loop 1 residues of subunits A-E (Fig. 4a). The two p97 rings are very similar to each other, and all of the structures were superimposed on the p97 D2 ring, which has slightly better density than the D1 ring. The pore loop 1 residues and bound substrate are very similar in all cases, with RMSD values ranging between 0.43-2.68 Å. By contrast, the superimposed structures show a range of positions for subunit F (Fig. 4b). The poor local resolution associated with these subunits is consistent with the model that subunit F is transitioning from the bottom to the top of the helical stack. The mechanistic implications of these observations are summarized in the Discussion section.

**Co-purifying adaptors and substrates**. p47 is associated with various cellular processes, including post-mitotic organelle membrane remodeling[24] and regulation of protein phosphatase 1 complex assembly[25], presumably by unfolding specific protein substrates. Consistent with this model, the presence of unfolded substrate density in our cryo-EM reconstruction indicates that p47 co-purifies with native substrates. To investigate the identity of this bound substrate (or collection of multiple substrates) p47 co-IPs were visualized by SDS-PAGE. This revealed several unidentified bands in addition to p97 (Supplementary Fig. 1B). In order to determine the identities of these co-purifying proteins, p47 co-IP eluates were subject to mass spectrometry proteomics (Supplementary Fig. 8). Proteins displaying strongly enriched peptides include myosin-9, insulin-degrading enzyme, and ornithine aminotransferase, which suggest they may be specific substrates of p47/p97. Interestingly, the yeast ortholog of ornithine aminotransferase, Car2p, was strongly enriched in pulldowns of yeast p47 (Shp1p)[18] and suggests their interactions are conserved from yeast to human. p47 is reported to associate with both non-ubiquitylated[25] and ubiquitylated substrates, with ubiquitylated forms including monoubiquitin[26] and K48-linked, and K63-linked chains[27,28]. Anti-ubiquitin immunoblot analysis revealed that polyubiquitin co-purifies with p47, and mass spectrometry indicated a strong enrichment of K48-linked peptides

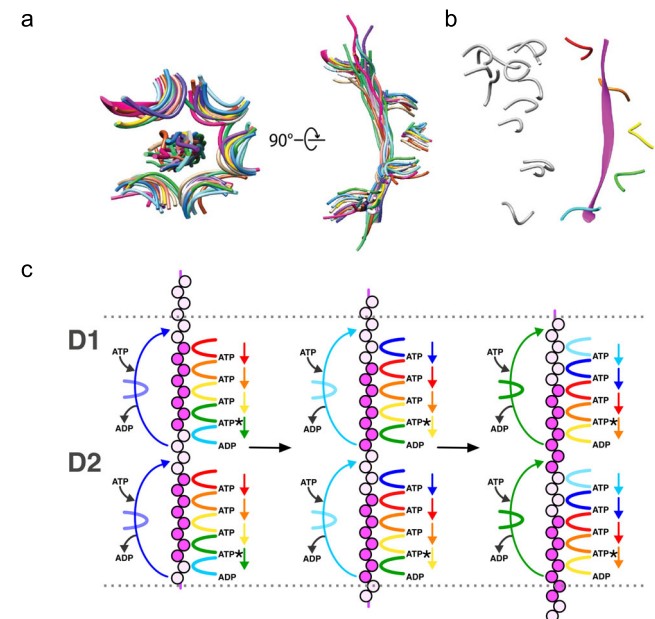

**Fig. 4 Substrate translocation model. a** p97 D2 pore loops (light purple) of subunits A-E superimposed with the equivalent residues from 14 AAA ATPases, including YME1 (PDB 6AZ0, tan)[12], Vps4 (6AP1, light blue)[14], D1 of p97 (this study, gray), Spastin (6PEN, pink)[20], TRIP13 (6F0X, magenta)[49], Msp1 (6PE0, yellow)[50], Katanin (6UGE, dark blue)[51], D1 and D2 of Cdc48 (6OPC, dark purple and purple, respectively)[18], Hsp104 (5VJH, dark green)[52], human 19 S regulatory particle (6MSE, turquoise)[53], yeast 19S regulatory particle (6EF3, brown)[54], Rix7 (6MAT, green)[55], and Abo1 (6JQ0, light green)[56]. **b** p97 D2 pore loops 1 (subunits A–E) and substrate displayed in colors. Subunit F pore loop 1 from the various AAA unfoldase structures shown in panel a displayed in gray. **c** Hand-over-hand translocation model. A strand of unfolding substrate (circles) is threaded through the central pore of the p97 hexamer. Adjacent pore loops (right arcs) form grooves that bind dipeptides of the unfolding substrate. ATP binding stabilizes the intersubunit interface of peptide-binding subunits. ATP hydrolysis and phosphate release (asterisks) weakens the interface and peptide binding. The sixth subunit (left arc) detaches from bottom of each hexamer stack. Nucleotide exchange and ATP binding to the detached subunit promotes its re-engagement to the next exposed dipeptide of the unfolding substrate and re-stabilizes the intersubunit interface. Colored arrows indicate directionality of movement for each subunit. The process repeats with the detachment of the bottom-most subunit and its re-binding at the top of each hexamer stack.

(Supplementary Fig. 1C, Supplementary Table 2). The presence of K6, K11, and K63 linkages were also detected, which suggests that p47 interacts with diverse ubiquitin chains. These findings are consistent with the model that substrate density in our reconstruction corresponds to the superposition of multiple different substrate proteins.

In addition to p47, p97 is regulated by many other binding partners that drive substrate recruitment, subcellular localization, and ATP hydrolysis rates[29,30]. Consistent with the model that many of these binding partners assemble in different combinations to regulate p97 activity, IP-MS of p47 pulldowns revealed an enrichment of other known p97 adaptors, including PLAA, UBXN1, UBXN7, and NPLOC4 (Supplementary Fig. 8). Notably, the extensively characterized Ufd1 adaptor protein was not enriched by this analysis. Like p47, many of the co-purifying adaptors feature UBX domains, which presumably bind to p97 N domains in the same manner as the p47 UBX. This suggests that the multiple N domains within one p97 hexamer may bind to different UBX domain adaptors. Curiously, we observed a long,

rod-shaped density that spans approximately 40 Å at the interface between subunits F and A (Supplementary Fig. 9). This density lies against the D1-D2 linker of subunit F and the large ATPase domain of the subunit A D1 ATPase cassette, but the source of the density is not apparent. Classification of particle images failed to improve the quality of this feature and did not reveal the presence of additional binding partners, which is not surprising given their sub-stoichiometric amounts detected by SDS-PAGE (Supplementary Fig. 1B). Nevertheless, the co-purification of multiple p97-binding partners suggests their interplay during substrate selection and processing.

## Discussion

Previously reported structures of p97 determined in the presence of various nucleotides showed the D1 and D2 rings in the same planar conformation as our inactive complex, and proposed models of the protein substrate unfolding mechanism have involving toggling between rotationally symmetric states that have all six subunits in either ATP or ADP-bound states[6]. In contrast, by using affinity-tagged p47 adaptor to isolate p97 from lysate in the presence of an appropriate nucleotide analog, we have visualized p97 in both the inactive rotationally symmetric state of earlier structures and bound with authentic substrate in an asymmetric, translocation-active conformation that resembles the substrate/peptide-bound structures reported in recent years for other AAA unfoldases[8].

The "down" position of N domains seen in our inactive rotationally symmetric state likely contribute to autoinhibition by providing additional interactions that stabilize a conformation that is unable to engage substrate[6]. This concept of an auto-inhibitory role for the N domain that is relieved upon substrate engagement is consistent with observations that disease-associated mutations that result in hyperactive p97 shift the "up"/"down" equilibrium toward the "up" position[31]. It is also consistent with the findings of a study published since submission of this manuscript[10], which also proposed that the planar six-fold rotationally symmetric state represents an inactive state and might be an attractive target for therapeutic p97 inhibitors.

Our asymmetric substrate-engaged structure shows density for substrate spiraling through the D1 and D2 pores in an extended conformation. This substrate lies in a peptide-binding channel that is formed by the pore loop 1 and pore loop 2 residues of both D1 and D2. The canonical beta-strand conformation of this binding channel is consistent with engagement of almost any polypeptide sequence. As with other recently reported AAA unfoldase structures, binding involves engagement of substrate side chains in a series of alternating pockets that are formed between the p97 pore loop residues of adjacent subunits. Five subunits form the substrate-binding groove, and are held in the helical configuration necessary for substrate engagement by binding of ADP·BeF$_x$ (ATP) at the subunit interfaces. The sixth subunit appears to be transitioning between the ends of the helix formed by the five spiraling subunits, and is disengaged from the substrate and makes less extensive interactions with its neighboring subunits.

Our substrate-engaged structure closely overlays with other peptide-bound AAA unfoldase complexes determined in recent years, including that of the yeast ortholog Cdc48 determined in the presence of authentic[18] or model[19] substrate. Cdc48/p97 is strongly conserved from yeast to human (69% sequence identity). Moreover, p47 and its yeast ortholog Shp1 are also conserved (29% identity). The close similarity between our current p97–p47-substrate structure and the previously reported Cdc48-Shp1-substrate structure[18] reinforces the conclusion that p97 employs the same mechanism of substrate engagement and

translocation as other AAA unfoldases. In this proposed mechanism (Fig. 4c), helical interfaces between the p97 subunits are stabilized by binding of ATP, ATP hydrolysis and phosphate release at the lowest interface weakens the interface to promote disengagement to the position of the transitioning subunit, and binding of ATP promotes formation of a new helical interface at the top of the helix and binding of the next substrate dipeptide. In this manner, p97 can walk along substrate two residues at a time, effectively pulling the substrate through the pore and enforcing an extended/unfolded conformation. Consistent with this general model for AAA unfoldase substrate translocation, superposition of multiple structures shows consistent conformations for the helical binding groove and engaged substrate, and a wide range of positions for the transitioning subunit (Fig. 4). The similarities in substrate unfolding by p97 between p47 and Ufd1-Npl4[10,19] also suggest a common mechanism among p97-binding partner complexes. We note, however, that that this mechanism differs from earlier proposals for p97[6,32] and is disputed for some other AAA unfoldases[33,34].

It is striking that the p97 D1 and D2 rings seem to be walking along the substrate in lock step, with the vertically aligned domains in each subunit at the same stage in the translocation cycle. The apparent coordination in conformation between the D1 and D2 rings implies that each step along a substrate dipeptide is twice as forceful as would be achieved by a single ring AAA unfoldase, such as the majority of those shown in the overlap of Fig. 4. Thus, p97 appears to operate in a low-gear, burning twice as much ATP as a single ring unfoldase for the same length of translocation. This may be important for the processing of more recalcitrant substrates, such as domains that are harder to unfold, are embedded in the lipid bilayer, or are aggregated. Consistent with this model, p97 is able to unfold substrates that are resistant to processing by other AAA+ ATPases, including the 26 S proteasome[35].

Our proteomic analysis indicated that multiple other adaptors copurify with the p47–p97 complex. Some of these adaptors probably bind to the same N domain site as p47, and may associate competitively either through equivalent UBX domains or through other interactions that obscure the same binding surface. An intriguing possibility is that the rod of density apparent at the interface between subunits A and F corresponds to an adaptor that can bind simultaneously with p47. However, the identity of protein giving rise to that density is not apparent, nor is it apparent how that association may change during the translocation cycle. The combinatorial diversity of p97-adaptor complexes and their effects on driving specific cellular processes will be a subject of future studies.

## Methods

**Cloning, protein expression, and purification of FLAG-tagged p47.** The expression constructs for His-p47-Flag were created by subcloning a 3xFLAG tag (amino acid sequence DYKDHDGDYKDHDIDYKDDDDK) to the 3′ end of the p47 coding sequence from the plasmid pTrcHis-p47 (a gift from Hemmo Meyer, Addgene plasmid #21268; https://n2t.net/addgene:21268; RRID: Addgene_21268)[36]. The modified construct was expressed in *Escherichia coli* BL21 (DE3) RIL cells (Stratagene) and grown in ZY autoinduction media at 37 °C for 3 h and then at 19 °C overnight[37]. Cells were pelleted by centrifugation, resuspended in lysis buffer (25 mM Tris-HCl pH 7.4, 450 mM NaCl, 20 mM imidazole, 1 mg/ml lysozyme, and a protease inhibitor cocktail comprising 0.5 µg/mL leupeptin, 0.5 µg/mL aprotinin, 0.7 µg/mL pepstatin, and 16.7 µg/mL PMSF), incubated for 45 min on ice, sonicated, and clarification by centrifugation (15,000 × g, 45 min, 4 °C). The supernatant was bound to nickel-NTA agarose (Qiagen) followed by washing with lysis buffer. Elution was performed with 300 mM imidazole in lysis buffer, followed by overnight dialysis at 4 °C in a buffer comprising 150 mM NaCl and 25 mM Tris-HCl pH 7.4. The dialyzed protein was further purified by anion exchange chromatography (Hi Trap Q HP 5 ml, Cytiva) over a gradient from 150 mM to 1 M NaCl in 25 mM Tris buffer at pH 7.4, and gel filtration chromatography in 100 mM NaCl and 20 mM HEPES buffer pH 7.4 (Hi Prep 16/60 Sephacryl S-200, Cytiva). Purified protein was aliquoted, snap frozen in liquid nitrogen, and stored at −80 °C.

**Cell culture**. HEK293GnTI$^{-/-}$ cells (ATCC CRL-3022) were grown in suspension in FreeStyle 293 Expression medium (Invitrogen, Carlsbad, CA) at 37 °C in an orbital shaker. Cells were treated using 100 nM bortezomib (CAS 179324-69-7, Millipore Sigma) when cell density reached ~3 ×10$^6$/ml. Cell pellets were harvested by centrifugation 4 h after bortezomib treatment, snap frozen in liquid nitrogen, and stored at −80 °C.

**Co-immunoprecipitation (co-IP)**. Co-IPs were performed using a rapid affinity purification strategy as previously described[18]. In brief, IP Buffer (100 mM KOAc, 10 mM MgCl$_2$, 25 mM HEPES-KOH pH 7.4, 1 mM ADP·BeF$_x$, 10% glycerol, 0.2% Igepal CA- 630, 1 mM DTT, and a protease inhibitor cocktail comprising 0.5 μg/mL leupeptin, 0.5 μg/mL aprotinin, 0.7 μg/mL pepstatin, and 16.67 μg/mL PMSF) was added to the frozen cell pellets at a 1:4 (w/v) ratio. Cells were lysed by dounce homogenization on ice. Lysates were clarified by centrifugation and supernatants used for co-IP experiments. Purified recombinant His-p47-FLAG (0.4 μM) was pre-incubated with equilibrated anti-FLAG M2 affinity gel (Sigma) for 30 min at 4 °C, followed by the addition of clarified lysate to the affinity gel. The affinity gel was washed extensively with IP Buffer, followed by several washes with IP buffer lacking detergent and glycerol. Resin-bound materials were eluted using synthetic 3xFLAG peptide (ApexBio) in the presence of 1 mM ADP·BeF$_x$ for 1 h at 4 °C, and eluted materials were recovered by pipetting. Eluted samples used for cryo-EM were crosslinked with a final concentration of 0.1% glutaraldehyde for 10 min at room temperature, quenched with Tris-HCl (pH 7.4), and then immediately used for vitrification. Non-crosslinked samples used for mass spectrometry were stored at −80 °C until further processing. Mock co-IPs as background controls for mass spectrometry analyses were performed as described above without the addition of His-p47-FLAG and either with or without the addition of ADP·BeF$_x$.

**Immunoblotting**. Samples were separated by SDS-PAGE, followed by electrophoretic transfer onto polyvinylidene difluoride membrane (Bio-Rad). Membranes were blocked with 5% nonfat milk in TBST (20 mM Tris-HCl, pH 8.0, 150 mM NaCl, 0.1% Tween-20) for 30 min at room temperature, and incubated for 1 h with the following antibodies: anti-ubiquitin clone E412J (rabbit monoclonal; 1:1,000; Cell Signaling Technology, 43124S) or anti-FLAG clone M2 (mouse monoclonal; Sigma-Aldrich, F1804). Membranes were washed in TBST, followed by incubation with secondary antibodies (goat anti-mouse IgG; 1:10,000; LI-COR, P/N 926-32210 and goat anti-rabbit IgG; P/N 926-68071) for 1 h at room temperature. Membranes were washed in TBST and digitized using the ODYSSEY CLx scanner (Source Data).

**Mass spectrometry**. Frozen eluates of p47 co-IP with ADP·BeF$_x$, mock co-IP with ADP·BeF$_x$, and mock co-IP without ADP·BeF$_x$ ($n = 3$ for each sample) were thawed with the addition of 50 μl of 5% SDS in 50 mM triethylamine bicarbonate pH 8.5 (TEAB) containing protease inhibitor cocktail (Sigma). Cysteines were reduced using 10 mM TCEP and alkylated using 40 mM chloroacetamide in a single incubation for 5 min at 100 °C. Samples were then sonicated in a bath sonicator for 5 min, collecting by centrifugation, acidified with 6 μl 12% phosphoric acid (H$_2$PO$_4$), and mixed briefly by vortexing. This solution was diluted with wash buffer (350 μL of 100 mM TEAB, 90% MeOH) at 4 °C, mixed briefly by vortexing, and placed in a suspension trap (STrap micro, Protifi, Farmingdale NY) by centrifuging at 4000 × $g$ for 30 s. The STrap was washed 3 times by centrifugation of 700 μL 50/50 IPA/MeOH through the STrap, followed by 3 washes of 400 μL of wash buffer. The sample was digested to peptides on the STrap column by addition of 40 μL of TEAB containing 1 μgram of trypsin (Sigma Aldrich sequencing grade) at 37 °C for 16 h. Sample was eluted into a clean collection tube with 3 sequential 50 μL washes (50 mM TEAB in ddH$_2$O, 0.2% formic acid in ddH$_2$O, 50% acetonitrile in ddH$_2$O) this solution was transferred to a MS sample vial, dried and resuspended in 10 μL of 0.1% formic acid 3% acetonitrile in water.

Mass spectrometry data were collected using an Orbitrap Fusion Lumos mass spectrometer (Thermo Fisher Scientific, Waltham, MA, USA) coupled to an EASY-nLC 1200 liquid chromatography (LC) pump (Thermo Fisher Scientific, Waltham, MA, USA). A capillary RSLC column (EASY-spray column pepMap RSLC, C18, 2 μm, 100 Å, 75 μm × 15 cm) was used for the separation of peptides. The mobile phase comprised buffer A (0.1% formic acid in optima water) and buffer B (optima water and 0.1% formic acid in 80% acetonitrile). Peptides were eluted at 300 nL/min with the following gradients over 2 h: 3–25% B for 80 min; 25–35% B for 20 min; 35–45% B for 8 min; 45–85% B for 2 min and 85% for 8 min. Data were acquired using the top speed method (3 s cycle). A full scan MS at a resolution of 120,000 at 200 *m/z* mass was acquired in the Orbitrap with a target value of 4e5 and a maximum injection time of 50 ms. Peptides with charge states of 2–6 were selected from the top abundance peaks by the quadrupole for collisional dissociation (CID with normalized energy 30) MS/MS, and the fragment ions were detected in the linear ion trap with target AGC value of 1e4, a maximum injection time of 35 ms, and a dynamic exclusion time of 60 s. Precursor ions with ambiguous charge states were not fragmented.

PEAKS Studio software (version X pro) was used for de novo sequencing and database searching to identify proteins in our raw MS data and to quantify, filter (quality-control), and normalize the quantitation data for each protein[38]. Peptides were identified from MS/MS spectra by searching against the Swiss-Prot human

proteome database (downloaded April of 2020) with a reverse sequence decoy database concatenated. Variables for the search were as follows: enzyme was set as trypsin with one missed cleavage site. Carbamidomethylation of cysteine was set as a fixed modification while N-terminal acetylation and methionine oxidation were set as variable modifications. A false-positive rate of 0.01 was required for peptides and proteins. Minimum length of peptide was set to 7 amino acids. At least 2 peptides were required for protein identification. The precursor mass error of 20 ppm was set for the precursor mass, and the mass error was set as 0.3 Da for the MSMS. Proteins identified from this first-level analysis of fragmentation spectra were used as a constrained database to search for ubiquitin modifications using the PTM module in PEAKS Studio. Label-free quantitation was enabled with MS1 tolerance ±20 ppm and a MS2 tolerance ±50 ppm, carbamidomethylation of cysteine was set as a fixed modification, while N-terminal acetylation and methionine oxidation were set as variable modifications. Peptide assignments with a false discovery rate less than 1% were included in comparative quantitative analyses and used to generate protein identification files for the quantitative and kinetic analyses. Relative concentrations of each protein were measured via label-free quantitation in the PEAKS software by normalizing the area under the curve (AUC) for each peptide and protein to the total ion count (TIC) in each sample. The probability of each protein being a p47 interactor was calculated by comparing 3 replicate samples from p47-FLAG co-IP eluates versus mock FLAG co-IP eluates using the SAINT software package[39]. The −logP values for each protein were calculated from P-values obtained using a two-way heteroscedastic t-test to compare the replicate measurements in each sample. No values were imputed for zeros, no multiple testing corrections were applied as the analysis relied on multiple criteria beyond p-value to establish significance.

**Electron cryo-microscopy**. UltrAuFoil R1.2/1.3 Au300 mesh grids (Quantifoil) were glow discharged for 1 min on each side at 25 mA using a Pelco easiGlow unit (Ted Pella, Inc.). 3.5 μl of purified sample were applied to the grid and blotted with filter paper (595 Filter Paper, Ted Pella, Inc.) for 2.5 s using a Mk. II Vitrobot (Thermo Fisher Scientific) with a −1 mm offset and then plunge frozen into liquid ethane.

Cryo-EM movies were recorded using SerialEM v3.8[40] in super-resolution mode on a 300 kV Titan Krios (Thermo Fisher Scientific) equipped with a post-GIF K3 direct detector (Gatan, Inc.). A total of 9732 movies were recorded at a nominal magnification of 81,000×, corresponding to a super-resolution pixel size of 0.54 Å, with a total dose of 46 electrons/Å$^2$ and 40 frames per movie.

**Image processing**. Super-resolution cryo-EM movie frames were motion corrected, dose weighted, Fourier-binned 2x, and summed using cryoSPARC Live as implemented in cryoSPARC v3.0[41]. CTF parameters were determined using patch CTF estimation. A total of 1,519,419 particles were selected across 9732 micrographs. Particles were extracted according to their unbinned physical pixel size (1.08 Å/px) and used for 2D classification using cryoSPARC, after which 457,076 particles were sorted into well-resolved 2D classes. A reconstruction of substrate-bound Cdc48 (EMD-20136)[18] was imported into cryoSPARC, low pass filtered to 40 Å, and used as the basis for heterogeneous refinement in cryoSPARC to sort particles into multiple classes. Resulting 3D classes revealed asymmetric, symmetric, and junk classes.

A total of 186,987 particles were classified into two classes with apparent C6 symmetry. Top views were dominant in these classes and contributed to preferred orientation artifacts. The particles were exported from cryoSPARC into RELION for further classification and auto-refinement. A final reconstruction of 5.7 Å resolution was achieved with 24,013 particles.

A total of 132,906 particles were classified into an asymmetric conformation. These were used for non-uniform refinement as implemented in cryoSPARC. This reconstruction led to a map at 3.8 Å resolution, after which further classification was performed using heterogeneous refinement with 2 classes. This yielded two classes, at 4.4 Å and 7.9 Å resolution. The 4.4 Å map comprised 85,965 particles, which were used for non-uniform refinement to achieve a final resolution of 3.6 Å.

All focused classification jobs were performed in RELION v3.1[42]. In brief, particle images were exported from cryoSPARC into RELION using pyem. Custom soft-edged masks were created in RELION (width_soft_edge = 6) over each p97(N)-p47(UBX) interface. Masks were applied in 3D classification jobs without particle alignment (K = 3). Following 3D classification, particles with UBX density were used for RELION 3D auto-refinement.

**Model building and refinement**. The structures of individual domains of each p97 subunit (PDB 5FTN)[6] were docked into the 3.6 Å map as rigid bodies using UCSF Chimera (version 1.15)[43]. Detailed model building was performed using Coot v0.8.7[44]. The substrate was built as an extended β-strand with residues 1–12 and 21–22 as alanine and residues 13–20 as alanine-valine repeats. D1 and D2 of subunits A to E and the substrate were subjected to real-space refinement using Phenix v1.19[45]. Secondary structure restraints were applied during refinements. Guided by visual inspection of map similarity, NCS restraints were applied to p97 subunits A-E with the exception of residues 332–340 of subunit A and residues 583–599 of subunits A–E. For subunits A, B, C, and D, the Be to ADP O3B distance was restrained to 1.6 Å, and the Mg$^{2+}$ to BeF$_3$ F1 distance was restrained to 2.0 Å.

The p97 N domain and p47 UBX domain complex structure (PDB 1S3S)[5] was docked into the unsharpened 3.6 Å map as a rigid body using UCSF Chimera. Molprobity v4.5[46] and EMRinger[47] as implemented in Phenix were used for additional structure validation. A summary of cryo-EM data collection, image processing, and model refinement statistics is provided in Table S1. All structural visualization figures were generated with UCSF Chimera[43].

**Reporting summary**. Further information on research design is available in the Nature Research Reporting Summary linked to this article.

## Data availability

All maps have been deposited to the EM Databank and are accessible via accession numbers EMD-23835 (substrate-bound p97–p47) and EMD-26654 (substrate-free p97–p47). The coordinate models are accessible on the Protein Data Bank via PDB ID 7MHS [https://www.rcsb.org/structure/7MHS] (substrate-bound p97–p47). The mass spectrometry proteomics data have been deposited to the ProteomeXchange Consortium via the PRIDE partner repository[48] with the dataset identifier PXD033451.

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

## Acknowledgements

This was work supported by grants to P.S.S. (NIH R35 GM133772), C.P.H. (grant NIH P50 AI150464), I.C. (NIH F31 CA254427), and J.C.P. (NIH R01 AG066874 and Fritz B. Burns Foundation). A portion of this research was supported by NIH grant U24GM129547 and performed at the PNCC at OHSU and accessed through EMSL, a DOE Office of Science User Facility sponsored by the Office of Biological and Environmental Research. We thank the University of Utah Arnold and Mabel Beckman Center for Cryo-EM and Center for High Performance Computing for cryo-EM and computational support, respectively.

## Author contributions

P.S.S. and C.P.H. conceived the study. P.S.S., Y.X., H.H., and J.C.P. designed the experiments. Y.X. and Y.G. performed biochemistry experiments. Y.X., H.H., I.C., and P.S.S. performed structural biology experiments. N.G.M., N.R.Z., and J.C.P. performed mass spectrometry experiments. All authors analyzed the results. P.S.S. and C.P.H. wrote the manuscript.

## Competing interests

The authors declare no competing interests.
