## [Peer Review File · Nature Communications]

Active conformation of the p97-p47 unfoldase complexReviewers' Comments:

Reviewer #1:

Remarks to the Author:

Xu et al. present structures of p97, an AAA+ unfoldase that plays a critical role in eukaryotic protein quality control, in complex with its adaptor protein, p47. The authors identify 2 coexisting states: a previously observed C6 symmetric, planar, substrate-free organization, and a substrate-bound, active state that is essentially identical to the spiraling organization seen for numerous other AAA+ proteins. These findings are important, because previous structures of p97 containing ATP did not have substrate threaded through the pore and did not present a spiraling organization (Banerjee et al Science, 2016). This led to speculation that p97 might deviate from the canonical hand-over-hand model for AAA+ activity that has emerged over the last 5 years. The data presented here support conservation of this fundamental mechanism of action in p97 and are consistent with recently published substrate-bound structures of the yeast homologue of p97, Cdc48 (Cooney et al Science, 2019, Twomey et al Science, 2019). These results are relevant and convincing, but the novelty is somewhat limited, and this study would require further structural analysis/discussion prior to publication in Nature Communications:

1. Given that the central conclusion of this manuscript is the conservation of the hand-over-hand mechanism powered by a sequential ATP hydrolysis cycle, the authors should add a panel in Figure 1 showing their density for the ADP bound nucleotide binding pocket to demonstrate that they can indeed unambiguously ascribe nucleotide state in their structure. Moreover, the authors should add supplementary panels showing a close up view of their density for the nucleotide binding pocket and the nucleotide (or absence thereof) for each subunit in both the D1 and D2 rings.
2. While the authors do not appear to overstate the conclusions that can be derived from their substrate-bound structure, additional processing approaches might help improve their resolution and further strengthen their claims. Have the authors attempted 3D classification without alignment or multibody refinement in relation to deal with the conformational heterogeneity that might currently be limiting their resolution? Could masking and focused classification improve the quality of their reconstruction for the individual D1 and D2 staircases (or subunits), as well as the N-terminal domains and p47? Did the authors explore the 3D variability analysis options in cryosparc?
3. The pore loop of the D1 ring of p97 does not contain the canonical aromatic residue critical for translocation in all other AAA+ proteins. The authors should mention this in the text. Also, could the authors comment on this substantial deviation from the norm?
4. Previous structures of the D1-D2 unfoldase Hsp104 showed that at the seam of the spiral D1-D1 and D2-D2 intersubunit interactions are disrupted and substituted by D1-D2 interactions (Gates et al, Science, 2017). These interactions were proposed to synchronize the ATP hydrolysis cycle in both rings such that it is offset by one subunit in the D2 ring (Deville et al Cell Rep, 2019). This inter-ring interactions do not appear to occur in p97 and D1 and D2 of a given subunit seem to fire simultaneously? Does the substrate-bound structure of p97 provide any insights into the allosteric mechanism(s) coordinating the sequential ATP hydrolysis cycle within each ring and between the rings? Does p97 contain an ISS motif as in YME1 and the 26S Proteasome (Puchades et al, Science 2017, de la Pena et al, Science 2018)? How is nucleotide state "communicated" to the neighboring subunits and the pore loops in the substrate-bound state of p97?
5. While the p97-p47 interaction has been the focus of previous crystallographic studies, the authors should discuss this interaction in the context of the spiraling, substrate bound organization in more detail. All 6 ATPases appear to be simultaneously bound to p47. Do the authors expect this to be the case under physiological conditions (with each p47 bringing a substrate molecule with it?) or could this be an artifact from the ADP-BeF substrate-trapped state? In the ADP-bound subunit, shouldn't the N-domain adopt a down conformation and drag p47 down with it?

6. The substrate-free structure presented here appears to suffer from pathological preferred orientation and the side view shown in the supplementary figure does not seem consistent with a 4.5Å structure. Anisotropic reconstructions typically lead to an overestimation of the resolution reported by FSC. The authors should clarify this in their manuscript and report the euler angle distribution as well as the 3D-FSC of this reconstruction in the supplementary figure.

7. Could the authors provide a more in-depth comparison between the previously solved cryo-EM substrate-free structure of p97 (Banerjee et al, Science, 2016) and the substrate-free structure they solved here? Can they ascribe nucleotide state and the location of the N-terminal domains in their structure? If so, are the nucleotide states they observe consistent with previous substrate-free structures? Also, could the authors comment on why/whether ATP binding is not directly linked to substrate binding and adoption of the spiraling organization in p97? Also, is p47 at all present in the symmetric, substrate-free state (perhaps in an ATP-dependent manner)?

Reviewer #2:

Remarks to the Author:

In this manuscript Xu and Han et al present the cryo-EM structure of p97 in a substrate engaged state. Analogous to other AAA-ATPases in the substrate engaged state of p97 adopts a spiral staircase like arrangement. Conserved pore loops lining the central channel grip the substrate to facilitate unfolding and support a hand-over-hand model of substrate unfolding. The structural work is well done and of interest given the significance of p97, however this manuscript is very brief and there is no biochemistry to support the structure or model. Moreover, given that there are already several published substrate engaged structures of the yeast homologue (Cdc48) the observation that p97 adopts the same configuration is not a significant finding on its own.

Concerns:

1. There are several substrate bound structures of Cdc48, the yeast homologue of p97, showing that Cdc48 adopts an asymmetric structure when it engages a substrate in the central channel (Twomey et al Science 2019, Cooney et al Science 2019). My major concern with this brief report is that Cdc48 is never mentioned in the main text. It's only mentioned as a superposition in the Fig. 2 legend. While it's very relevant to solve the active structure of human p97, the authors must put this into better context by describing what's already been demonstrated with Cdc48 and other closely related AAA-ATPases such as VAT.
2. The p97-p47 complex was isolated by co-IP using bacterial expressed p47 as bait. The eluate shown in Fig. S1 appears to be full of lots of proteins suggesting that the sample is highly heterogenous. Has this been analyzed by mass spec to determine what is in this sample? Were the authors able to pull out any additional 2D classes of other p97-p47 complexes during 2D classification?
3. What is the substrate in the central channel? I suggest the authors use mass-spec and/or cross-linking approaches to determine the identity of the substrate(s) in the central channel.
4. The resolution of the NTD and p47 is poor, which is not surprising given that the NTD of p97 is mobile. Have the authors tried 3D variability or multi-body refinement to look at the conformational landscape of these regions?
5. Does the substrate bound structure of p97 reveal any new insight into the known p97 disease causing mutations?
6. There is virtually no discussion of p47 in this manuscript. Was any new information gained about the structure of p47 when bound to p97 in the asymmetric state?

Reviewer #3:

Remarks to the Author:

The manuscript presents an interesting structure of an important AAA ATPase p97 in its substrate-bound conformation. Distinct from the previous 6-fold rotational symmetric structures of p97 observed in the absence of substrates, the substrate-bound p97 shows a helical arrangement of its subunits, which is consistent with a number of other related AAA ATPases, suggesting that p97 also employs the same 'hand-over-hand' translocation and unfolding mechanism previously postulated for other AAA unfoldases. However, in my opinion, there is limited new information that this study presents to the p97 community, considering the recently published work on Cdc48, the yeast homology of p97, on the same topic from the same group. The structure of the substrate-bound p97, the arrangement of the figures, the points of discussion, and the conclusion are, in my humble opinion, all very similar to the previous Cdc48 paper (Cooney et al., *Science* 365, 502–505 (2019)). In addition, the 'hand-over-hand' translocation mechanism is currently a widely accepted mechanism for this class of AAA unfoldases. Although the author mentioned a possible alternative mechanism of p97 in which the ATPase processes the substrate 'by toggling between the up/down state', it is not considered the most probable hypothesis especially due to a few solid studies in the recent years where pulling of substrates into the central pore has been observed (*Cell* 169, 722–735, May 4, 2017; *Molecular Cell* 72, 1–12, November 15, 2018). In this sense, the study here confirmed with yet another case what seems to be a highly speculated general mechanism for this family of enzymes without giving enough detailed/new understandings which are specific to p97. Therefore, in my opinion, I do not recommend this work to be published in *Nature Communication* in its current form.

In addition, there are specific points I would like to suggest.

1. The writing of the paper could use substantial polishing with detailed discussion of the structural features and insights provided by the structure into the function of p97.
2. Is there any method one can use to identify the substrate observed in the cryo-EM structure?
3. Can the authors provide comparison between this substrate-bound p97 structure with the previous published substrate-bound Cdc48 structure (Cooney et al., *Science* 365, 502–505 (2019)), as well as with the Cdc48 structure with poly-ubiquitinated substrate and UN cofactor (E. C. Twomey et al., *Science* 10.1126/science.aax1033 (2019))?
4. References seem to be rather limited and incomplete with a total of 9 references in the main text. More original and seminal studies on structure and mechanism of p97 should be included.
5. In the sentence "The other asymmetric class displayed five of the six p97 protomers in a helical arrangement that is superimposable with structures of other substrate-bound AAA unfoldases (Fig. 1B, C)7." Reference 7 is incorrectly referenced.
6. The fragmented densities in the vicinity of the N-domain have been assigned to UBX of p47. Has there been any experimental evidence confirming this assignment? Can the authors provide an explanation to the apparent feature in figure 1e where the densities of the UBX domains are mostly observed with the three N-domains from one side of the complex?
7. The authors commented that 'all six N domains appear to be bound to UBX domain'. However, considering the low densities of N-domain and the highly fragmented densities of UBX domains, judged from Figure 1e, is it possible that each N domain is occupied by the UBX domain substoichiometrically and the densities of the UBX domain are averaged from all particles? A recent study on the interaction between p97 and p47 has shown that a deep classification of the dataset of the complex reveals various binding configurations (FigS12 in <https://doi.org/10.1073/pnas.2013920117>)

8. Figure 2f is not the most intuitive in order to visualize the 'hand-over-hand' mechanism. The dash lines of the loops on the left could be mistakenly interpreted as if it involves five discreet steps for the bottom subunit to move to the top. In addition, the meaning of the four colored arrows on the right is not very clear. More descriptive figure legend is needed to help the readers to understand the illustration better.

We appreciate the reviewers' comments. We have revised the manuscript from a Brief Communications to an Article format. Reviewer comments are copied verbatim. Our point-by-point responses are provided below in blue text:

REVIEWER COMMENTS

Reviewer #1 (Remarks to the Author):

Xu et al. present structures of p97, an AAA+ unfoldase that plays a critical role in eukaryotic protein quality control, in complex with its adaptor protein, p47. The authors identify 2 coexisting states: a previously observed C6 symmetric, planar, substrate-free organization, and a substrate-bound, active state that is essentially identical to the spiraling organization seen for numerous other AAA+ proteins. These findings are important, because previous structures of p97 containing ATP did not have substrate threaded through the pore and did not present a spiraling organization (Banerjee et al Science, 2016). This led to speculation that p97 might deviate from the canonical hand-over-hand model for AAA+ activity that has emerged over the last 5 years. The data presented here support conservation of this fundamental mechanism of action in p97 and are consistent with recently published substrate-bound structures of the yeast homologue of p97, Cdc48 (Cooney et al Science, 2019, Twomey et al Science, 2019). These results are relevant and convincing, but the novelty is somewhat limited, and this study would require further structural analysis/discussion prior to publication in Nature Communications:

We appreciate the reviewer's comments that the results are relevant and convincing. The additional analysis and discussion as requested are described below.

1. Given that the central conclusion of this manuscript is the conservation of the hand-over-hand mechanism powered by a sequential ATP hydrolysis cycle, the authors should add a panel in Figure 1 showing their density for the ADP bound nucleotide binding pocket to demonstrate that they can indeed unambiguously ascribe nucleotide state in their structure. Moreover, the authors should add supplementary panels showing a close up view of their density for the nucleotide binding pocket and the nucleotide (or absence thereof) for each subunit in both the D1 and D2 rings.

As requested, our revised manuscript includes a new figure that shows nucleotide density and model for the first five interfaces (Figure 2). A figure that shows surrounding density and model for all six interfaces is also included as a new supplementary figure (Extended Data Fig. 5).

2. While the authors do not appear to overstate the conclusions that can be derived from their substrate-bound structure, additional processing approaches might help improve their resolution and further strengthen their claims. Have the authors attempted 3D classification without alignment or multibody refinement in relation to deal with the conformational heterogeneity that might currently be limiting their resolution?

We have extended the analysis of our dataset by performing 3D variability analysis in cryoSPARC and focused 3D classification in RELION. Variability is not apparent in the data, i.e. cryoSPARC did not detect significant variability and, as described below, focused 3D classification in RELION over the D1, D2 rings, and subunit F, did not yield multiple classes.

Could masking and focused classification improve the quality of their reconstruction for the individual D1 and D2 staircases (or subunits), as well as the N-terminal domains and p47?

Focused classification over D1 or D2 domains did not improve reconstruction quality. In contrast, focused classification over the N-domains revealed variable p47 occupancies for both the inactive (down) and substrate-bound (up) conformations. We have updated the main text to describe these results as follows:

For the substrate-free “down” conformation: “The periphery of the N-domains contains additional densities that are consistent with the UBX domain of p47 (Dreveny et al. 2004) (Extended Data Fig. 3F). Focused classification over each of the six N-domains revealed UBX occupancies between 59-93%, suggesting that most, but not all, of the potential UBX-binding sites are engaged with the adaptor (Extended Data Fig. 3G,H). Although the N-domains are not saturated with UBX binding, we cannot rule out the possibility that some p47 may have dissociated from p97 between protein isolation and cryo-EM grid preparation. Nevertheless, our results are consistent with a recent report that reconstituted p47-p97 complexes display variable levels of UBX occupancy (Conicella et al. 2020).”

For the substrate-bound “up” conformation: “Focused 3D classification over individual N-domains revealed UBX occupancies of 14-22% (Extended Data Fig. 7). The discrepancy in UBX occupancies between the ‘up’ and ‘down’ conformations may reflect different stoichiometries of p47 binding when the complex is active or idle. We also note the possibility that the increased mobility of the N-domain in the ‘up’ position may weaken potential UBX densities and reduce the sensitivity of focused 3D classification.”

Did the authors explore the 3D variability analysis options in cryosparc?

Yes, as mentioned above, 3D variability analysis did not reveal regions of notable variability.

3. The pore loop of the D1 ring of p97 does not contain the canonical aromatic residue critical for translocation in all other AAA+ proteins. The authors should mention this in the text. Also, could the authors comment on this substantial deviation from the norm?

The text has been updated to describe this observation as follows: “Interestingly, substrate density is weaker in D1 than it is in D2, which implies a weaker interaction with the D1 pore loops. This is consistent with the identity of D2 residues Trp551 and Phe552, which seem ideal for the formation deep notches that, in the highly solvated context of the p97 pore, can accommodate essentially any substrate side chain. By contrast, the equivalent D1 residues, Leu278 and Ala279, create shallower binding pockets for the substrate side chains. Consequently, consistent with the weaker density, substrate is likely to be bound less tightly in D1 compared to D2. Weaker interactions in D1 appear to be conserved between yeast and human18, and to be important for biological function because substitutions of the orthologous Leu278 to Phe, Trp, or Tyr causes a lethal phenotype in yeast (Esaki et al., 2017).”

4. Previous structures of the D1-D2 unfoldase Hsp104 showed that at the seam of the spiral D1-D1 and D2-D2 intersubunit interactions are disrupted and substituted by D1-D2 interactions (Gates et al, Science, 2017). These interactions were proposed to synchronize the ATP hydrolysis cycle in both rings such that it is offset by one subunit in the D2 ring (Deville et al Cell Rep, 2019). This inter-ring interactions do not appear to occur in p97 and D1 and D2 of a given subunit seem to fire simultaneously? Does the substrate-bound structure of p97 provide any

insights into the allosteric mechanism(s) coordinating the sequential ATP hydrolysis cycle within each ring and between the rings?

Does p97 contain an ISS motif as in YME1 and the 26S Proteasome (Puchades et al, Science 2017, de la Pena et al, Science 2018)? How is nucleotide state “communicated” to the neighboring subunits and the pore loops in the substrate-bound state of p97?

p97/Cdc48 and Hsp104 belong to different clades of AAA+ ATPases and have different D1-D2 interactions. To illustrate this point, we attach the following side-by-side view of Hsp104 (left, D1 yellow; D2 orange) and p97 (right, D1 cyan; D2 dark blue) protomers. In p97, the D1-D2 linker (green) mediates inter-ring interactions through the formation of extensive contacts with both D1 and D2 residues. The D1-D2 linker in Hsp104 (magenta) is differently oriented and does not make such interactions. A detailed comparison between p97 and Hsp104 is beyond the scope of the paper and we feel that it is unnecessary to include it in the revised manuscript.

Our structures of yeast and human p97 support a model in which D1 and D2 subunits fire simultaneously. The mechanism may be supported through the ISS motif. We now include a brief description of the ISS in the manuscript along with a new supplementary figure as follows:

“Like other members of the classical AAA clade of proteins, p97 contains a conserved intersubunit signaling (ISS) motif, which has been proposed to transmit information about the nucleotide status between adjacent subunits and to the pore loops (Puchades et al., *Science* 2017). Specifically, D1 L335 and D2 M611, which appear to be the key residues from the ISS in each domain, interact with the hydrophobic residues near the Walker B motifs of the neighboring subunit (Extended Data Fig. 6). This ISS structure is well conserved in the subunits that form a helix in the asymmetric substrate-engaged reconstruction of p97, and are closely superimposable with the equivalent regions from YME1, Vps4, Cdc48, and human Spastin, but are somewhat divergent in *Drosophila* Spastin.”

5. While the p97-p47 interaction has been the focus of previous crystallographic studies, the authors should discuss this interaction in the context of the spiraling, substrate bound organization in more detail. All 6 ATPases appear to be simultaneously bound to p47. Do the authors expect this to be the case under physiological conditions (with each p47 bringing a substrate molecule with it?) or could this be an artifact from the ADP-BeF substrate-trapped state?

As described above, focused classification reveals 14-22% of individual N-domains contain robust p47-UBX density. This suggests that not all subunits are simultaneously bound to p47.

The exact stoichiometry of p47 on substrate-free and substrate-bound p97 hexamers will be a focus of future studies.

In the ADP-bound subunit, shouldn't the N-domain adopt a down conformation and drag p47 down with it?

The N-domain of the ADP-bound subunit (subunit E) adopts an 'up' conformation in our substrate-bound reconstruction. We believe the down conformation will be favored when the hexamer adopts the inactive planar conformation that does not bind substrate and likely binds ADP at all subunits. This is because the down conformation is stabilized by interactions that are unique to the planar state. Thus, we do not expect the N domain of one subunit to be down just because that subunit is bound to ADP.

6. The substrate-free structure presented here appears to suffer from pathological preferred orientation and the side view shown in the supplementary figure does not seem consistent with a 4.5Å structure. Anisotropic reconstructions typically lead to an overestimation of the resolution reported by FSC. The authors should clarify this in their manuscript and report the euler angle distribution as well as the 3D-FSC of this reconstruction in the supplementary figure.

The substrate-free structure was reprocessed to address orientation bias. A lower resolution was achieved (5.7 Å) due to fewer particles used (i.e., particles from over-represented views were removed), but with more isotropic directional FSCs. The validation data, including orientation distribution and 3D FSC plots, are now included as a new supplementary figure (Extended Data Fig. 3).

7. Could the authors provide a more in-depth comparison between the previously solved cryo-EM substrate-free structure of p97 (Banerjee et al, *Science*, 2016) and the substrate-free structure they solved here?

The relatively low resolution of our substrate-free reconstruction does not permit model refinement. We therefore limit our analysis of the model to rigid-body fitting of existing substrate-free structures, which agrees with structures of substrate-free p97 in an ADP-bound state (Banerjee et al., *Science* 2016) and a structure of p47(UBX)-p97(N-D1) (Dreveny et al., *EMBO J.* 2004). We have updated the main text to include these details as follows:

"The symmetric reconstruction does not display bound substrate and presumably represents an idle state of the complex. In this conformation, both D1 and D2 rings are stacked in planar arrangements. The N-domains adopt a 'down' conformation that is co-planar with the D1 ring. This is consistent with existing structures of p97 in an ADP-bound state (Banerjee et al., *Science* 2016), although the resolution of our reconstruction does not permit nucleotide assignment. The periphery of the N-domains contains additional densities that are consistent with the UBX domain of p47 (Dreveny et al., *EMBO J.* 2004) (Extended Data Fig. 3F)."

Can they ascribe nucleotide state and the location of the N-terminal domains in their structure? If so, are the nucleotide states they observe consistent with previous substrate-free structures?

The resolution of our substrate-free structure does not permit assignment of nucleotide states. The N-terminal domains are exclusively in the 'down' position as reported in other inactive structures.

Also, could the authors comment on why/whether ATP binding is not directly linked to substrate binding and adoption of the spiraling organization in p97?

We believe that ATP binding is directly linked to substrate binding because it stabilizes the spiral conformation that displays the substrate peptide-binding groove. We have tried to make this model clear in the revised manuscript.

Also, is p47 at all present in the symmetric, substrate-free state (perhaps in an ATP-dependent manner?)?

As described above, focused 3D classification was used to explore p47 occupancy over each N-domain in the symmetric state and confirmed that most available binding sites are bound by the UBX domain.

Reviewer #2 (Remarks to the Author):

In this manuscript Xu and Han et al present the cryo-EM structure of p97 in a substrate engaged state. Analogous to other AAA-ATPases in the substrate engaged state of p97 adopts a spiral staircase like arrangement. Conserved pore loops lining the central channel grip the substrate to facilitate unfolding and support a hand-over-hand model of substrate unfolding. The structural work is well done and of interest given the significance of p97, however this manuscript is very brief and there is no biochemistry to support the structure or model. Moreover, given that there are already several published substrate engaged structures of the yeast homologue (Cdc48) the observation that p97 adopts the same configuration is not a significant finding on its own.

We appreciate the reviewer's comments that the work is well done and of interest. As requested, the revised manuscript has been expanded to include new insights into p97-p47 substrates and additional binding partners. The structural similarities between various Cdc48- and p97-adaptor complexes and other AAA+ ATPases reinforce a growing body of evidence that the mechanism of substrate unfolding is unified and deeply conserved.

Concerns:

1. There are several substrate bound structures of Cdc48, the yeast homologue of p97, showing that Cdc48 adopts an asymmetric structure when it engages a substrate in the central channel (Twomey et al Science 2019, Cooney et al Science 2019). My major concern with this brief report is that Cdc48 is never mentioned in the main text. It's only mentioned as a superposition in the Fig. 2 legend. While it's very relevant to solve the active structure of human p97, the authors must put this into better context by describing what's already been demonstrated with Cdc48 and other closely related AAA-ATPases such as VAT.

Our revised paper now references relevant publications throughout the manuscript, including Twomey et al. (2019), Cooney et al. (2019), Ripstein et al. (2017), and the mutant substrate-bound p97 structure that was published after our initial submission (Pan et al., 2021).

2. The p97-p47 complex was isolated by co-IP using bacterial expressed p47 as bait. The eluate shown in Fig. S1 appears to be full of lots of proteins suggesting that the sample is highly heterogenous. Has this been analyzed by mass spec to determine what is in this sample?

Our revised manuscript now includes detailed mass spectrometry proteomics analysis of p47 co-IPs. These analyses reveal an enrichment of other p97 binding partners, ubiquitin signatures,

and other putative substrates compared to controls. The results are described in the main text (last two paragraphs of Results section), a new figure (Extended Data Fig. 8), and a new table (Supplementary Table 2). Confirmation of ubiquitin enrichment in our p47 pulldowns is also now shown in Extended Data Fig. 1C.

Were the authors able to pull out any additional 2D classes of other p97-p47 complexes during 2D classification?

No, all well-resolved 2D classes were pooled for downstream 3D classification and refinement.

3. What is the substrate in the central channel? I suggest the authors use mass-spec and/or cross-linking approaches to determine the identity of the substrate(s) in the central channel.

As suggested, and as described above, mass spectrometry reveals potential substrates that are described in the revised manuscript.

4. The resolution of the NTD and p47 is poor, which is not surprising given that the NTD of p97 is mobile. Have the authors tried 3D variability or multi-body refinement to look at the conformational landscape of these regions?

As noted above in our response to Reviewer 1, we have applied 3D variability analysis and focused 3D classification to explore the conformational and compositional heterogeneity of the N-domains. Results are described above and included in the main text.

5. Does the substrate bound structure of p97 reveal any new insight into the known p97 disease causing mutations?

Most pathogenic p97 mutations are clustered in regions of poor local resolution in our reconstruction (e.g., N-domains and N-D1 linker). Implications for the mechanistic impact of disease-causing mutations is discussed in a paragraph that has been included in the Discussion section of our revised manuscript.

6. There is virtually no discussion of p47 in this manuscript. Was any new information gained about the structure of p47 when bound to p97 in the asymmetric state?

No new information about the structure of p47 was gained from our study. The poor local resolution of the N-domain only permits rigid-body fitting of the p47-UBX domain into our reconstruction. Other functional domains, i.e. UBA, SEP, and SHP-box, are predicted to be separated from the UBX domain by flexible linkers.

Reviewer #3 (Remarks to the Author):

The manuscript presents an interesting structure of an important AAA ATPase p97 in its substrate-bound conformation. Distinct from the previous 6-fold rotational symmetric structures of p97 observed in the absence of substrates, the substrate-bound p97 shows a helical arrangement of its subunits, which is consistent with a number of other related AAA ATPases, suggesting that p97 also employs the same 'hand-over-hand' translocation and unfolding mechanism previously postulated for other AAA unfoldases. However, in my opinion, there is limited new information that this study presents to the p97 community, considering the recently published work on Cdc48, the yeast homology of p97, on the same topic from the same group. The structure of the substrate-bound p97, the arrangement of the figures, the points of

discussion, and the conclusion are, in my humble opinion, all very similar to the previous Cdc48 paper (Cooney et al., Science 365, 502–505 (2019)). In addition, the 'hand-over-hand' translocation mechanism is currently a widely accepted mechanism for this class of AAA unfoldases. Although the author mentioned a possible alternative mechanism of p97 in which the ATPase processes the substrate 'by toggling between the up/down state', it is not considered the most probable hypothesis especially due to a few solid studies in the recent years where pulling of substrates into the central pore has been observed (Cell 169, 722–735, May 4, 2017; Molecular Cell 72, 1–12, November 15, 2018). In this sense, the study here confirmed with yet another case what seems to be a highly speculated general mechanism for this family of enzymes without giving enough detailed/new understandings which are specific to p97. Therefore, in my opinion, I do not recommend this work to be published in Nature Communication in its current form.

We agree that AAA+ ATPases use a general mechanism of substrate translocation, but the details are still unsettled in the field. The mechanistic discrepancies are now acknowledged in our Discussion section, along with commentary on the implications of p97 function as a double-ringed unfoldase.

In addition, there are specific points I would like to suggest.

1. The writing of the paper could use substantial polishing with detailed discussion of the structural features and insights provided by the structure into the function of p97.

The revised manuscript now includes discussion of new insights learned from extended image processing (p97-p47 interactions via the p47 N-domain), mass spectrometry (putative substrates and additional p97 binding partners), and more detail about structural features in comparison with other AAA+ ATPases.

2. Is there any method one can use to identify the substrate observed in the cryo-EM structure?

As described in our response to Reviewer 2's comments, we have applied mass spectrometry to p47 co-IPs and include results and discussion in our revised manuscript.

3. Can the authors provide comparison between this substrate-bound p97 structure with the previous published substrate-bound Cdc48 structure (Cooney et al., Science 365, 502–505 (2019)), as well as with the Cdc48 structure with poly-ubiquitinated substrate and UN cofactor (E. C. Twomey et al., Science 10.1126/science.aax1033 (2019))?

Comparative analyses and a discussion of mechanistic implications are summarized in the Discussion section.

4. References seem to be rather limited and incomplete with a total of 9 references in the main text. More original and seminal studies on structure and mechanism of p97 should be included.

The manuscript now includes additional references.

5. In the sentence "The other asymmetric class displayed five of the six p97 protomers in a helical arrangement that is superimposable with structures of other substrate-bound AAA unfoldases (Fig. 1B, C)7." Reference 7 is incorrectly referenced.

Thank you for pointing out this error, which has been fixed.

6. The fragmented densities in the vicinity of the N-domain have been assigned to UBX of p47. Has there been any experimental evidence confirming this assignment? Can the authors provide an explanation to the apparent feature in figure 1e where the densities of the UBX domains are mostly observed with the three N-domains from one side of the complex?

The focused classification over p97 N-domains as described above improves the quality of the p47 UBX domains and permits reliable rigid-body fitting of existing UBX crystal structures, as described in our response to Reviewer #1 (point #7). Our reprocessed data show more consistent UBX density across the well-resolved protomers (i.e., subunits A-E).

7. The authors commented that 'all six N domains appear to be bound to UBX domain'. However, considering the low densities of N-domain and the highly fragmented densities of UBX domains, judged from Figure 1e, is it possible that each N domain is occupied by the UBX domain substoichiometrically and the densities of the UBX domain are averaged from all particles? A recent study on the interaction between p97 and p47 has shown that a deep classification of the dataset of the complex reveals various binding configurations (FigS12 in <https://doi.org/10.1073/pnas.2013920117>)

We agree that the binding is likely to be substoichiometric. As described in our response to Reviewer 1, focused 3D classification was used to determine UBX occupancy over each protomer for both inactive and substrate-bound conformations. The following sentence and citation were added to accompany the description of our new results: "Nevertheless, our results are consistent with a recent report that reconstituted p47-p97 complexes display variable levels of UBX occupancy (Conicella et al. 2020)."

8. Figure 2f is not the most intuitive in order to visualize the 'hand-over-hand' mechanism. The dash lines of the loops on the left could be mistakenly interpreted as if it involves five discreet steps for the bottom subunit to move to the top. In addition, the meaning of the four colored arrows on the right is not very clear. More descriptive figure legend is needed to help the readers to understand the illustration better.

A new model figure has been created along with a more descriptive figure legend (Fig. 4C).

Reviewers' Comments:

Reviewer #1:

Remarks to the Author:

All my concerns were appropriately addressed by the authors

Reviewer #2:

Remarks to the Author:

In this revised manuscript Xu and Han present a cryo-EM structure of active p97 in a substrate engaged state. The structure supports that p97 like many other AAA-ATPases uses a processive hand-over-hand mechanism to drive substrate translocation. In the revision the authors have expanded upon the limited text from the initial submission and nicely compared p97 with many other AAA-ATPases. The additional mass-spectrometry results also provide new insight into p97 binding partners and ubiquitin signatures. While the overall manuscript has been substantially improved, I still find the overall significance is somewhat limited.

Minor Concerns:

- Fig. 4A – Please define the colors in the figure legend.
- Fig. S5 – Please color the AB, BC, etc labels as in Fig. 2
- Fig. S6 – Please provide an overlay of the electron density for the ISS motif

Reviewer #3:

Remarks to the Author:

The authors have effectively addressed the comments and answered the questions raised in the last round of the reviewing process. I recommend the work be published.

We appreciate the reviewers' comments, which are copied verbatim below in black text. Our point-by-point responses are provided below in blue text:

REVIEWERS' COMMENTS

Reviewer #1 (Remarks to the Author):

All my concerns were appropriately addressed by the authors

We thank the reviewer for evaluating the manuscript.

Reviewer #2 (Remarks to the Author):

In this revised manuscript Xu and Han present a cryo-EM structure of active p97 in a substrate engaged state. The structure supports that p97 like many other AAA-ATPases uses a processive hand-over-hand mechanism to drive substrate translocation. In the revision the authors have expanded upon the limited text from the initial submission and nicely compared p97 with many other AAA-ATPases. The additional mass-spectrometry results also provide new insight into p97 binding partners and ubiquitin signatures. While the overall manuscript has been substantially improved, I still find the overall significance is somewhat limited.

We thank the reviewer for evaluating the manuscript.

Minor Concerns:

- Fig. 4A – Please define the colors in the figure legend.

Colors are now defined in the figure legend for Fig. 4A.

- Fig. S5 – Please color the AB, BC, etc labels as in Fig. 2

Text labels in Supplementary Fig. 5 are now colored as in Fig. 2.

- Fig. S6 – Please provide an overlay of the electron density for the ISS motif

An overlay of the electron density for the ISS motif is now provided in Supplementary Fig. 6.

Reviewer #3 (Remarks to the Author):

The authors have effectively addressed the comments and answered the questions raised in the last round of the reviewing process. I recommend the work be published.

We thank the reviewer for evaluating the manuscript.